# Furan-Conjugated Tripeptides as Potent Antitumor Drugs

**DOI:** 10.3390/biom10121684

**Published:** 2020-12-16

**Authors:** Hunain Ali, Almas Jabeen, Rukesh Maharjan, Muhammad Nadeem-ul-Haque, Husena Aamra, Salma Nazir, Serab Khan, Hamza Olleik, Marc Maresca, Farzana Shaheen

**Affiliations:** 1International Center for Chemical and Biological Sciences, H. E. J. Research Institute of Chemistry, University of Karachi, Karachi 75270, Pakistan; hunain200195@gmail.com (H.A.); rukeshmaharjan2013@gmail.com (R.M.); nadeem.and.chem@gmail.com (M.N.-u.-H.); husenaaamra@gmail.com (H.A.); salmasiddiqi2020@gmail.com (S.N.); serab305@gmail.com (S.K.); 2Dr. Panjwani Center for Molecular Medicine and Drug Research, International Center for Chemical and Biological Sciences, University of Karachi, Karachi 75270, Pakistan; almasjabeen.aj@gmail.com; 3Aix Marseille University, CNRS, Centrale Marseille, iSm2, 13397 Marseille, France; hamza.olleik@live.com; 4Department of Biology, American University of Beirut, Beirut 1107 2020, Lebanon

**Keywords:** peptide conjugates, furan-based tripeptides, heterocycles, Rink amide resin, HeLa cell line, anticancer

## Abstract

Cervical cancer is among the leading causes of death in women. Chemotherapy options available for cervical cancer include highly cytotoxic drugs such as taxol, cisplatin, 5-florouracil, and doxorubicin, which are not specific. In the current study, we have identified a new peptide conjugate (Fur^4^-2-Nal^3^-Ala^2^-Phe^1^-CONH_2_) (conjugate **4**), from screening of a small library of tripeptide-conjugates of furan, as highly potent anticancer compound against human cervical cancer cells (HeLa cells) (IC_50_ = 0.15 ± 0.05 µg/mL or 0.28 +/− 0.09 µM). Peptides were constructed on Rink amide resin from *C*- to *N*-terminus followed by capping by α-furoic acid moiety. The synthesized peptides were purified by recycling RP-HPLC, and structures of all the peptides were confirmed by using FABMS/ESIMS, ^1^H- NMR, ^13^C-NMR, and HR-FABMS. Conjugate **4** was furthermore found to be specifically active against human cervical cancer cells since it did not inhibit the proliferation of other human normal cells (HUVEC (human umbilical vein endothelial cells) and IMR-90 (normal human fibroblasts)), and cancer cells tested (HUVEC, MCF-7, and MDA-MB-231 cells), as well as in mice 3T3 cells (normal fibroblasts). This study revealed a good structure activity relationship of various peptide conjugates. Conjugate **4** in branched forms (**4a** and **4b**) were also synthesized and evaluated against HeLa cells, and results revealed that both were inactive. Atomic force microscopy (AFM) studies and staining with rhodamine 123 and propidium iodide (PI) revealed that conjugate **4** possesses a membranolytic effect and causes the loss of mitochondrial membrane potential.

## 1. Introduction

Cervical cancer is the fourth most common type of cancer among women [1]. In lower- and middle-income communities (LMIC), it is second most common type of cancer among women, overall it is the third most common cancer after oral and breast cancer. It is responsible for 7.5% of deaths among women due to cancer, out of which 90% of cases were reported from LIMC [2]. Due to lack of knowledge about its prevention, screening, and vaccination programs, around 70% of these cancer cases are diagnosed at advanced stages which results in a higher mortality rate [3]. Treatments available for cervical or any other types of cancers are invasive and are usually not specific. For example, bleomycin, a modified glycopeptide-based drug used for the treatment of cervical cancer, is highly toxic against cancerous cells but has several side effects and is also highly toxic against normal cells [4,5].

Peptides are important therapeutic agents due to their lower toxicity and higher specificity [6]. They possess many biological benefits and are preferable drug candidates [7]. Around 60 FDA-approved peptide-based drugs are commercially available, while hundreds of them are also at different development stages. It is estimated that 75% of peptide-based medicinal agents are used in the treatment of cancer [8]. Their selective nature makes them a promising drug candidate [9]. Peptides identified from different natural sources have poor solubility, low bioavailability, and are sensitive to proteolytic enzymes [10,11,12]. These limitations are overcome by introducing some changes to the skeleton of the peptide chain, such as the introduction of various unusual amino acids or conjugation with heterocyclic moieties. Such peptidomimetics are reported to be more stable, as well as having enhanced potency when compared to classical structure of peptides [12].

Peptide conjugates are diverse molecules possessing great therapeutic potential. In comparison to small molecules, peptides-based drugs are more specific and less toxic, and the conjugation of such peptides with small medicinally important molecules has been reported to enhance the selectivity of the molecules [9] Conjugation of small peptides with various anticancer drugs have been found to have enhanced anticancer potential [13]. The conjugation of paclitaxel with nonadecapeptide angiopep-2 has now gone into Phase II of clinical trials as a cure for brain cancer [14].

Heterocycles conjugated with peptide chains of different lengths are a novel class of lead molecules which have greater advantages in the field of drug discovery and development. It has been reported that selectivity, solubility, high cell permeability, and stability of the compounds improved after conjugation with amino acids. It has also been reported that the conjugation of a biologically active heterocyclic moiety in peptide chains enhances the potency of the compounds by increasing cell permeability [9,15]. Peptide chains such as GVGVP, VGVP, and GVP, conjugated with quinazolinone nucleus, were reported to enhance antibacterial activities [16].

Various drugs containing furan moiety are available commercially, such as non-steroidal anti-inflammatory drugs refecoxib and firocorib [17,18], and anti-microbial drugs such as diloxanide, cefuroxime, and ceftiofur [19].

Peptide-conjugated heterocycles represent an important class of medicinal agents [20]. Recently, a short peptide sequence library of compounds based on 2-amino-3-(3′,4′,5′-trimethoxybenzoyl)benzo[b]furan scaffold was reported to have very good therapeutic potential against HeLa cells and some other cancer cell lines [21]. Tripeptides conjugated with a furoyl moiety at C-terminal were reported to inhibit protease enzymes with greater selectivity and enhanced effectiveness and also possessed antineoplastic activity [22]. Furan-capped tripeptides containing unusual residues such as 4-hydroxyphenylglycine were reported to exhibit protease inhibition activity against dengue virus (DENV) and West Nile virus (WNV) [23]. Several studies have reported that peptides with D-amino acids or non-natural side-chain-variant amino acids (peptidomimetics), and blocked N- and C-termini through cyclization, have increased rigidity that results in reduced sensitivity to proteases [24,25,26]. In view of the aforementioned, the current study was designed on the synthesis and anticancer screening of a furoyl moiety conjugated with a tripeptide sequence containing natural, unnatural, hydrophobic, and cationic amino acid residues, in order to develop novel conjugates with enhanced protease stability and to check their potential against human cervical cancer cells.

## 2. Materials and Methods

### 2.1. Materials

All the chemicals used in the experiment portion were 95–99% pure. Fmoc-protected Rink amide resin, with loading capacity 0.602 mmol/g and mesh size 200–300, was used as a solid support. All the protected amino acids and resin were purchased from Chem. Impex (Wood Dale, IL, USA). All the solvents used are of HPLC grade. International Bruker Avance NMR spectrometer (600 MHz) was used to record NMR of samples. QSTAR XL MS/MS SYSTEMS (AB Science, Chatham, NJ, USA) and a JEOL-600H direct probe were used to record ESI-MS and FAB-MS spectra, respectively. Purification and percent purity determination were done using RP-HPLC and UPLC on a recycling preparative HPLC (LC-908W) (Japan Analytical Industry Co., Ltd. Tokyo, Japan) Jaigel ODS-MAT 80 (C18) column with an ACN:H_2_O:TFA (60:40:0.08) solvent system. Melting points were recorded using glass capillaries on Yanaco MP-S3 apparatus. A Schimadzu 1800 spectrophotometer was used to record UV (ultraviolet) absorption spectra, while s JASCO 302-A Infrared Spectrometer was used to record IR (infrared) absorption spectra.

### 2.2. General Experimental Procedure for Synthesis of Furan-Conjugated Tripeptides

Rink amide resin (0.5 g, 0.6 mmol) was swelled in 15 mL DMF for 1 h. After soaking, the Fmoc group was removed using 5 mL of 20% 4-methylpiperidine solution in DMF for 30 min. Amino acids (five equivalents) were loaded on pre-swollen resin with the help of DIC (five equivalent) and Oxymapure (five equivalent) in DMF. First coupling was done twice for 24 h, to make sure the resin was fully saturated, while next consequent couplings were done each for 3 h until the desired length was obtained. The Fmoc group was removed after each coupling step mentioned earlier. Resin was washed thrice with 10 mL of DMF, DCM, and DMF after each successive step. After the completion of the reaction, the peptidyl resin was cleaved using 20 mL of a variously composed TFA cocktail for 2 h. After 2 h the resin was filtered and washed with 10 mL of the TFA cocktail, along with washing the filtrate was concentrated under vacuum pressure and precipitation was performed with cold diethyl ether. The precipitates were washed with cold diethyl ether and used for further analysis after purification on RP-HPLC (Figure 1). UPLC profiles, NMR, and mass spectra of conjugates are shown in Appendix A.

**Conjugate 1:** Purple amorphous solid, 82.06 mg (51.7%); mp, 214 °C; UV (ACN) λ_max_ (log *ε*) 252 (3.85) nm; IR (KBr, cm^−1^) 3299 (NH), 3071 (=C-H), 2974 (C-H), 1645 (-NHC=O stretching), 1543 (Aromatic -C=C-, CN stretching, NH bending), 1424 (C-H bending, Aromatic -C=C- stretching), and 1208 (C-O stretching, CN stretching, NH bending); ^1^H-NMR (DMSO-d_6_, 600 MHz) and ^13^C-NMR (DMSO-d_6_, 150 MHz) see Appendix A.**Conjugate 2:** White amorphous solid, 50.04 mg (31.5%); mp 212–213 °C; UV (ACN) λ_max_ (log *ε*) 252 (4.15) nm; IR (KBr, cm^−1^) 3265 (NH), 3058 (=C-H), 2931 (C-H), 1641 (-NHC=O stretching), 1533 (Aromatic -C=C-, CN stretching, NH bending), 1471 (C-H bending, Aromatic -C=C- stretching), and 1201 (C-O stretching, CN stretching, NH bending); ^1^H-NMR (DMSO-d6, 600 MHz) and ^13^C-NMR (DMSO-d_6_, 150 MHz) see Appendix A.**Conjugate 3:** White amorphous solid, 81.5 mg (60.0%); mp 212–213 °C; UV (ACN) λ_max_ (log *ε*) 250 (4.14) nm; IR (KBr, cm^−1^) 3300 (NH), 30,791 (=C-H), 3011 (C-H), 1633 (-NHC=O stretching), 1546 (Aromatic -C=C-, CN stretching, NH bending), 1423 (C-H bending, Aromatic -C=C- stretching), and 1218 (C-O stretching, CN stretching, NH bending); ^1^H-NMR (DMSO-d_6_, 600 MHz) and ^13^C-NMR (DMSO-d_6_, 150 MHz) see Appendix A.**Conjugate 4:** Pale yellow amorphous solid, 80.17 mg (50.5%); mp, 173 °C; UV (ACN) λ_max_ (log *ε*) 225 (4.12) and 254 (3.42) nm; IR (KBr, cm^−1^) 3299 (NH), 3086 (=C-H), 2972 (C-H), 1643 (-NHC=O stretching), 1544 (Aromatic -C=C-, CN stretching, NH bending), and 1215 (C-O stretching, CN stretching, NH bending); ^1^H-NMR (DMSO-d_6_, 600 MHz) and ^13^C-NMR (DMSO-d_6_, 150 MHz) see Appendix A.**Conjugate 5:** Purple amorphous solid, 82.99 mg (47.7%); mp, 205 °C; UV (ACN) λ_max_ (log *ε*) 256 (3.62) nm; IR (KBr, cm^−1^) 3303 (NH), 3075 (=C-H), 2989 (C-H), 1680, 1649 (-NHC=O stretching), 1544 (Aromatic -C=C-, CN stretching, NH bending), 1433 (C-H bending, Aromatic -C=C- stretching), and 1213 (C-O stretching, CN stretching, NH bending); ^1^H-NMR (DMSO-d_6_, 600 MHz) and ^13^C-NMR (DMSO-d_6_, 150 MHz) see Appendix A.**Conjugate 6:** Purple amorphous solid, 69.4 mg (43.7%); mp 199-206 °C; UV (ACN) λ_max_ (log *ε*) 252 (4.18) nm; IR (KBr, cm^−1^) 3298 (NH), 3068 (=C-H), 2958 (C-H), 1680, 1645 (-NHC=O stretching), 1543 (Aromatic -C=C-, CN stretching, NH bending), 1436 (C-H bending, Aromatic -C=C- stretching), and 1214 (C-O stretching, CN stretching, NH bending); ^1^H-NMR (DMSO-d_6_, 600 MHz) and ^13^C-NMR (DMSO-d_6_, 150 MHz) see Appendix A.**Conjugate 7:** White amorphous solid, 60.3 mg (37.9%); mp, 135 °C; UV (ACN) λ_max_ (log *ε*) 256 (4.04) nm; IR (KBr, cm^−1^) 3294 (NH), 3057 (=C-H), 2917 (C-H), 1670, 1633 (-NHC=O stretching), and 1528 (Aromatic -C=C-, CN stretching, NH bending); ^1^H-NMR (DMSO-d_6_, 600 MHz) and ^13^C-NMR (DMSO-d_6_, 150 MHz) see Appendix A.**Conjugate 8:** White amorphous solid, 51.56 mg (32.4%); mp, 237 °C; UV (ACN) λ_max_ (log *ε*) 252 (4.09) nm; IR (KBr, cm^−1^) 3296 (NH), 3075 (=C-H), 2950 (C-H), 1677, 1643 (-NHC=O stretching), 1546 (Aromatic -C=C-, CN stretching, NH bending), 1426 (C-H bending, Aromatic -C=C- stretching), and 1227 (C-O stretching, CN stretching, NH bending); ^1^H-NMR (DMSO-d_6_, 600 MHz) and ^13^C-NMR (DMSO-d_6_, 150 MHz) see Appendix A.**Conjugate 9:** Grey amorphous solid, 85.95 mg (52.5%); mp 194–199 °C; UV (ACN) λ_max_ (log *ε*) 254 (4.24) nm; IR (KBr, cm^−1^) 3398 (NH), 3304 (OH), 3094 (=C-H), 2989 (C-H), 1676, 1650 (-NHC=O stretching), 1538 (Aromatic -C=C-, CN stretching, NH bending), 1431 (C-H bending, Aromatic -C=C- stretching), and 1212 (C-O stretching, CN stretching, NH bending); _1_H-NMR (DMSO-d_6_, 600 MHz) and ^13^C-NMR (DMSO-d_6_, 150 MHz) see Appendix A.**Conjugate 10:** Brown amorphous solid, 72.43 mg (46.8%); mp, 220 °C; UV (ACN) λ_max_ (log *ε*) 249 (4.07) nm; IR (KBr, cm^−1^) 3301 (NH), 3055 (=C-H), 2935 (C-H), 1634 (-NHC=O stretching), 1512 (Aromatic -C=C-, CN stretching, NH bending), 1452 (C-H bending, Aromatic -C=C- stretching), and 1204 (C-O stretching, CN stretching, NH bending); ^1^H-NMR (DMSO-d_6_, 600 MHz) and ^13^C-NMR (DMSO-d_6_, 150 MHz) see Appendix A.

### 2.3. Anti-Proliferative Assay

The anti-proliferative effect of peptides was tested using various normal and cancer cells. Human cancer cells used were human cervical cancer cells HeLa (ATCC^®^ CCL-2), and human breast cancer cells MCF-7 (ATCC^®^ HTB22) and MDA-MB-231 (ATCC^®^ HTB-26). Normal cells used were NIH-3T3 (ATCC^®^ CRL-1658), human normal endothelial cells (human umbilical vein endothelial cells, HUVEC) (ECACC, Sigma-Aldrich, Lyon, France), and human lung fibroblasts IMR-90 (ATCC^®^ CCL186). NIH-3T3, HeLa, IMR-90, MDA-MB-231, and MCF-7 cells were grown in DMEM supplemented with 10% fetal bovine serum (FBS), 1% L-glutamine, and 1% antibiotics (all from Invitrogen (Carlsbad, CA, USA)). HUVEC cells were cultured in specific medium (endothelial cell growth medium from Sigma Aldrich, Lyon, France). Cells were routinely grown on 25 cm^2^ flasks. For anti-proliferative assay, cells were detached from flasks using a trypsin-EDTA solution (from Thermofisher, Illkirch-Graffenstaden, France). After counting using a Malassez chamber, cells were diluted in appropriate culture media and seeded into 96-well cell culture plates (Greiner bio-one, Paris, France) at approximately 6000 cells/well. Cells were then treated with increasing concentration of conjugates diluted in appropriate medium. Doxorubicin was used as positive control. After 48 h medium was aspirated and the number of viable cells was colorimetrically measured as previously described [27,28,29]. Briefly, after exposure to increasing concentrations of conjugates or doxorubicin for 48 h, cell viability was measured using the standard MTT or resazurin assay. For the MTT assay, the supernatant was discarded and 50 µL of 0.5 mg/mL MTT dye was then added to each well and plates were further incubated for 4 h at 37 °C. The MTT was aspirated and 100 µL of DMSO was then added to each well. The extent of MTT reduction to formazan in cells was calculated by measuring the absorbance at 540 nm, using a spectrophotometer (Spectra Max plus, Molecular Devices, CA, USA). For the resazurin assay, the supernatant was discarded and 100 µL of resazurin solution (from Sigma Aldrich, Lyon, France) was added to each well. After 4 h of incubation at 37 °C, fluorescence associated to resazurin biotransformation by live cells was measured using spectrofluorometer (excitation 530 nm/emission 590 nm). In both cases, the antiproliferative activity was recorded as concentration causing 50% growth inhibition (IC_50_). Experiment was performed in triplicate (*n* = 3).

### 2.4. Atomic Force Microscopy of HeLa Cells

HeLa cells were grown in Dulbecco’s modified eagle medium (DMEM) supplemented with 10% fetal bovine serum. After 70–80% confluency was achieved, cells were treated with conjugate **4** at 20 µg/mL and incubated for 24 h at 37 °C in a 5% CO_2_ incubator. The next day, these cells were visualized using a microscope. In the untreated control, cells were healthy and attached, therefore, the cells were trypsinized to detach the cells from the wells, and then collected in a sterile Eppendorf tube. In conjugate **4** treated wells, cells were dead and detached. These dead cells were freely floating and therefore the cells were collected without using the trypsinization process in an Eppendorf tube. These cells collected in Eppendorf tubes were washed twice with sterile distilled water and then 10 µL of suspension was deposited into mica chips coated with 0.01% poly-L-lysine, to allow the cells attachment. This cell suspension was then dried at room temperature. Soon after, these mica chips were inserted inside an Agilent Technologies 5500 microscope and the chips were scanned at tapping mode with a silicon nitride cantilever. After finding cells in the chips, these cells were further scanned under a higher magnification. Atomic force microscopy (AFM) topography images were obtained, and we processed them in pseudo color images to differentiate the images according to the height. PicoView 1.2 software was used to process and analyze these images.

### 2.5. Rhodamine 123 and Propidium Iodide Staining Assay

As mentioned above, HeLa cells were grown in DMEM medium and seeded 3000 cells/well into 96-well plates. A stock solution of rhodamine 123 (Sigma-Aldrich, St Louis, MO, USA) and prodidium iodide (Sigma-Aldrich, St Louis, MO, USA) at concentrations of 10 and 1 mg/mL were prepared in pure DMSO and water, respectively.

The next day, 1 mL of the fresh media with 2% FBS was made, in which rhodamine 123 and propidium iodide dyes were added to make final concentrations of 10 and 5 µg/mL, respectively. Compounds **4**, **4a**, and **4b** at four different concentrations (10, 20, 30, and 40 µg/mL) were mixed with 1 mL of this media. Old media from the plates were aspirated from each well and 200 µL of fresh media containing compounds, rhodamine 123, and PI dye were added in their respective wells in triplicate and incubated at 37 °C in a 5% CO_2_ incubator. After 1 and 2 h incubation, media was aspirated from the wells and washed with PBS. These cells were visualized for rhodamine 123 and PI dye staining in a Nikon Eclipse TE2000-U fluorescence microscope at 200× magnification in FITC and TXR filters respectively. Images of cells at the same location for both dyes were captured. Rhodamine 123 and PI staining give green and red fluorescence, respectively. These two different colored images were then merged with one another using ImageJ software (NIH, Bethesda, MD, USA).

### 2.6. Synthesis of Dendrimeric Conjugate ***4a***

Jeffamine linker (2.4 g, 3 eq) was loaded on pre-swell into deprotected Rink amide resin (1.0 g, 0.602 mmol/g), and this step was repeated after 24 h with one equivalent of Jeffamine T-403 linker and three equivalents each of DIC and Oxymapure. After the deprotection of the Fmoc group, the conjugate **4** was constructed in a similar manner as described earlier, with three equivalents each of acid, DIC and OxymaPure for 3 h. Cleavage was done using a TFA cocktail composed of TFA:H_2_O (95:05). The peptide was purified using normal phase preparative TLC, using a solvent system DCM:MeOH (9.5:0.5) with few drops of acetic acid. Dimeric conjugate **4a** was confirmed by ESIMS (Appendix A).

### 2.7. Synthesis of Dendrimeric Conjugate ***4b***

Fmoc-Lysine (Alloc)-OH (1.36 g, 5 eq) was loaded on pre-swell into deprotected Rink amide resin (1.0 g, 0.602 mmol/g), and this step was repeated after 24 h with three equivalents each of Fmoc-Lysine (Alloc)-OH, DIC, and Oxymapure. The deprotection of the Fmoc group was done as described earlier. The conjugate **4** was constructed in a similar manner as described with three equivalents each of acid, DIC, and OxymaPure for 3 h. Cleavage was done using a TFA cocktail composed up of TFA:H_2_O (95:5) (Appendix A).

## 3. Results

In the current study, initially a twenty-member furan-conjugated tripeptide library was synthesized using Fmoc synthetic methodology on Rink amide resin, in which each member had randomly selected L- or D-configured natural (hydrophobic, hydrophilic, and cationic) and unnatural amino acid residues (2-l-Nal, 2-d-Nal, Phg, N-Me-l-Ala) (Appendix A). All compounds were fully characterized by mass spectrometry and detailed NMR spectroscopic studies (Appendix A).

The antiproliferative activity of conjugate **4** was then further evaluated using various normal and cancer cell types (Table 2). Interestingly, conjugate **4** was found to be inactive against human breast cancer cells (MCF-7 and MDA-MB-231 cells) and on normal cells (HUVEC (human umbilical vein endothelial cells), IMR-90 (normal human fibroblasts), and 3T3 cells (mice normal fibroblasts)). So far, the obtained results showed a selective anti-proliferative effect of conjugate **4** against human cervical cancer cells HeLa, contrary to doxorubicin which inhibits the proliferation of all cell types tested in this study (Table 2).

Conjugate **4** was identified as the most active compound in the series of conjugates **1–10** with an IC_50_ of 0.15 +/− 0.05 µg/mL or 0.28 +/− 0.09 µM. According to literature reports, linear peptides are classically found to be less active when compared to branched short peptides [30]. Two dimeric forms of peptide conjugate **4** were thus designed and synthesized to check the effect of the dimeric form of conjugate **4** on its activity on HeLa cells.

The Jeffamine-based linker was synthesized as reported in the literature, using Jeffamin T-403 as a starting material [31]. Two dendrimeric forms of conjugates (**4a** and **4b**) were prepared by linking each pair of monomeric conjugate **4** to a shared Lys residue or Jeffamine linker at the *C*-terminal of the conjugate (Scheme 1). The antiproliferative effects of the Jeffamine-based dendrimeric conjugate **4a** and the lysine-based dendrimeric conjugate **4b** were evaluated against HeLa cells and they were found to be inactive (IC_50_ > 100 µg/mL). These results show that conjugate **4** requires the C-terminus free in the form of amide for its activity against HeLa cells.

### Mechanistic Studies

After evaluation of the anticancer potential of conjugate **4**, further studies were focused on the mechanistic effect of this compound. As reported in the literature, the atomic force microscopy technique is used to investigate the mechanism of action of anticancer peptide against HeLa cell lines [32]. HeLa cells were visualized through atomic force microscopy, and the AFM images of untreated and treated cells were presented in Figure 2. Untreated control cells were detached from the wells by using a trypsinization process, and therefore anticipated cells were not in their exact morphology as detachment caused by trypsin shrunk the cells into a circular or oval shape. Control cells were intact with a smooth surface, and no signs of any cytoplasmic leakage. The background was clear showing no debris of cells. Cells treated with conjugate **4** at 20 µg/mL for 24 h showed scattered cytoplasmic content around cells. This cytoplasmic content is released from cells where the cell membrane has been ruptured due to the membranolytic effect of conjugate **4**.

To further confirm the membranolytic effect of conjugate **4**, rhodamine 123 and propidium iodide (PI) staining assay was performed. HeLa cells were treated with conjugate **4** at 20, 30, and 40 µg/mL. Rhodamine 123 stain, PI stain, and merged images are presented in Figure 3. Upon observation, untreated HeLa cells were found to be intact with a prominent nucleus and elongated filamentous branches. In fluorescein isothiocyanate (FITC) channel observation, untreated cells showed a high green fluorescence due to the binding of lipophilic cationic rhodamine 123 dye into the mitochondria. This shows untreated cells maintained highly negative mitochondrial transmembrane potential (MMP). Conversely, PI dye didn’t show fluorescence as it is impermeable to intact membrane cells. When cells were treated with increasing concentrations of conjugate **4**, there was a significant decrease in rhodamine fluorescence due to the dissipation of MMP. At the higher concentration tested (i.e., 40 µg/mL), 1 or 2 h treatment with conjugate **4** totally suppressed rhodamine fluorescence showing the complete dissipation of MMP of these cells. In contrast, PI dye fluorescence increased due to the penetration of dye through ruptured cell membranes, caused by the membrane disrupting effect of conjugate **4** in accordance with the AFM result. These PI-stained cells started to shrink and attained a circular shape due to loss of filamentous branches. Upon merging both the rhodamine 123 and PI images, there was only red fluorescence detected. Importantly, cells treated with inactive conjugates **4a** and **4b** only showed green fluorescence of rhodamine without PI fluorescence, confirming that these compounds do not affect MMP or the membrane integrity of HeLa cells (data not shown). In overall, rhodamine 123 and PI assay confirmed the fact that the anticancer effect of conjugate **4** is related to the alteration of mitochondria and permeabilization of the cell membrane of HeLa cells, these effects being observed within the first hours of exposure.

## 4. Discussion

Furan-conjugated tripeptides were synthesized using Fmoc-assisted solid-phase peptide methodology and were evaluated for their anticancer potential on HeLa cells. We have observed that all of the conjugates which contained least one of the D-amino acids (**11–20**, Appendix A) were inactive, while all conjugates with amino acids in L-configuration possess varying degrees of anticancer activity (Table 1). It was determined that the conjugate **2** (2-Fur^4^-L-2-Nal^3^-L-Phe^2^-L-Ala^1^-CONH_2_) with all amino acids in L-configuration was active (IC_50_ = 4.4 ± 0.2 µg/mL), while, the conjugate **1** (2-Fur^4^-D-2-Nal^3^-D-Phe^2^-D-Ala^1^-CONH_2_) with the same amino acids in D-configuration was inactive (IC_50_ >100 µg/mL). A good SAR of conjugate **2** helped us to optimize its sequence in the form of the most active conjugate **4** from this study (Figure 4). In the active sequence of conjugate **2**, first of all, when the Phe^2^ of conjugate **2** was replaced with different amino acid residues, the activity changes from strongly active to weakly active in comparison with the original, as can be seen in conjugate **3** (Phe^2^→Ala^2^, IC_50_ = 21.1±0.6 µg/mL), conjugate **5** (Phe^2^→2-Nal^2^, IC_50_ = 2.59 ± 0.3 µg/mL), conjugate **9** (Phe^2^→Tyr^2^, IC_50_ = 19.2 ± 3.2 µg/mL), and conjugate **10** (Phe^2^→Phg^2^, IC_50_ = 4.7 ± 0.6 µg/mL). It was observed that changing the Phe^2^ residue with Ala^2^ (as in conjugate **3**) decreased the activity by fivefold (IC_50_ = 21.1±0.6 µg/mL), while when replacing with Phe^2^ with 2-Nal (as in conjugate **5**), the activity increases twofold (IC_50_ = 2.59 ± 0.3 µg/mL). Shuffling the position of Phe^2^ with Ala^3^ of conjugate **2** has resulted in significantly enhanced activity (as observed in conjugate **4** with an IC_50_ value of 0.15 ± 0.05 µg/mL). It was observed that only those peptides in which Phe is present adjacent to L-alanine are strongly to moderately active. Thus conjugate **4** (IC_50_ = 0.15 ± 0.05 µg/mL), was identified as the most active compound in a series of compounds **1–10**. It was found that conjugate **4** was selective in nature as it only inhibits HeLa cancer cells with no effect on MCF-7, MDA-MB-231, HUVEC, IMR-90, and 3T3 cells.

The most active compound, conjugate **4**, was further linked with lysine and Jeffamine linker in order to check the effect of branching on the activity of conjugate **4**, and it was found that the C-terminal amide bond is important in the activity of conjugate 4 as branched conjugates 4a and 4b are inactive. The mechanism of action of conjugate **4** was studied using AFM and it showed a membranolytic affect, which was further confirmed using rhodamine 123 and PI staining assay. Although it needs to be further evaluated, these results suggest that conjugate **4** interacts with membrane lipids and inserts into them, leading to pore formation and membrane disruption. The reason for the selective action of conjugate **4** against cervical cancer cells will need also to be further explored.

## 5. Conclusions

In this study, a series of furan-conjugated tripeptides was synthesized and evaluated against human cervical cancer cells using HeLa cells as a model. The structure activity relationship was established in the peptide library. Although different conjugates gave interesting inhibitory activity against HeLa cells, conjugate **4** was identified as the most active with an IC_50_ of 0.15 ± 0.05 µg/mL. Importantly, the use of various normal and cancer cell lines allowed us to demonstrate that the anti-proliferative effect of conjugate **4** is selective of cervical cancer cells since it was found to be inactive on other cell types tested. Finally, atomic force analysis suggested that the mechanism of action of conjugate **4** on cervical cancer cells relies on the membranolytic effect and alteration of mitochondria. Further studies will be performed to characterize the precise mechanism of action of conjugate **4** on the membrane of cervical cancer cells in order to understand the reason for its selectivity of this cancer type.

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
