# Peer review of "Furan-Conjugated Tripeptides as Potent Antitumor Drugs"

_biomolecules, 2020, doi:10.3390/biom10121684_

Round 1
Reviewer 1 Report
The main aims of this manuscript is to study the anti-cancer potential of 10 tripeptide-conjugates of furan. The results showed that conjugate 4 (Fur4-2-Nal3-Ala2-Phe1-CONH2) exhibited the stronger anti-cancer effect against human cervical cancer HeLa cells, but not other cells including human normal IMR-90 and cancer HUVEC, MCF-7 and MDA-MB-231 cells. The anti-cancer potential of conjugates 4a and 4b derived from conjugate 4 were decreased since the C-terminal were linked with Lys residue or Jeffamine linker.
There are some comments as below.
- Scheme 1 and table 1 can be merged into one figure.
- The IC50 results in table 1 can be moved to table 2.
- Tables 3 and 4 should be as the supplementary results.
- Please remove the description “as well as mice normal 3T3 fibroblasts cells” in Abstract, because the manuscript don’t show the results in Table 5. However, the author still can mention no effect of conjugate 4 against 3T3 cells in the text.
- Figures 1 and 2 can be merged into one figure.
- In Figure 4, please check the period of conjugate 4 treatment at 40 μg/mL. 1 or 2 hrs?
- Since the conjugates 4a and 4b exhibited no effect against HeLa cells, Figure 5 can be removed. However, the description can be kept in the text.
- Figure 6 is very interesting. Please give more discussion the relation between the anti-cancer potentials and the chemical structures of these conjugates.
- Line 290, “In FITC channel observation, untreated cells showed highly green fluorescence due to binding of lipophilic cationic rhodamine 123 dye into the mitochondria. This shows untreated cells maintained highly negative mitochondrial transmembrane potential (MMP)”. The phenomenon isn’t shown in Figure 4. And it is necessary to show the cells without fluorescence of Rhodamine 123 or PI staining.
Author Response
Dear Reviewer,
We thank you for your constructive comments and suggestions. We tried our best to answer to all your questions. Please find below.
|
S. No. |
Comments |
Answers |
|
1 |
Scheme 1 and table 1 can be merged into one figure. |
Scheme 1 and table 1 are now merged into one figure in the revised manuscript. |
|
2 |
The IC50 results in table 1 can be moved to table 2. |
The IC50 results in table 1 are moved to table 2. |
|
3 |
Tables 3 and 4 shouldbe as the supplementary results. |
Tables 3 and 4 are removed from the main manuscript and now transferred to supplementary results pages 5 and 6 |
|
4 |
Please remove the description “as well as mice normal 3T3 fibroblasts cells” in Abstract, because the manuscript don’t show the results in Table 5. However, the author still can mention no effect of conjugate 4 against 3T3 cells in the text. |
The description of mice normal 3T3 fibroblasts cells” is removed from the Abstract. Results of 3T3 have been included in the manuscript in table 3. |
|
5 |
Figures 1 and 2 can be merged into one figure. |
No need anymore as figures have been modified according to reviewers suggestions |
|
6 |
In Figure 4, please check the period of conjugate 4 treatment at 40 μg/mL. 1 or 2 hrs? |
Figure 4 is correct, the period of conjugate 4 treatment at 40 μg/mL was 1 hr |
|
7 |
Since the conjugates 4a and 4b exhibited no effect against HeLa cells, Figure 5 canberemoved. However, the description canbekept in the text. |
Figure 5 is removed from the manuscript |
|
8 |
Figure 6 is very interesting. Please give more discussion the relation between the anti-cancer potentials and the chemical structures of these conjugates. |
The SAR has been discussed in the revised version, lines 312 to 331 |
|
9 |
Line 290, “In FITC channel observation, untreated cells showed highly green fluorescence due to binding of lipophilic cationic rhodamine 123 dye into the mitochondria. This shows untreated cells maintained highly negative mitochondrial transmembrane potential (MMP)”. The phenomenon isn’t shown in Figure 4. And it is necessary to show the cells without fluorescence of Rhodamine 123 or PI staining. |
We thanks the reviewer for this question. Please find further explanations regarding this assay. As rhodamine 123 gives green fluorescence therefore, it was visualized in FITC channel. Live viable cells maintain high negative mitochondrial transmembrane potential due to which rhodamine 123 dye penetrate inside and bind with mitochondria and give bright green fluorescence. These cells were seen with well-developed branches. In live cells, membrane is intact therefore PI dye will not penetrate inside the cell therefore no red fluorescence seen. Upon treatment, apoptosis process started due to which mitochondrial transmembrane dissipated and caused less binding of rhodamine 123. These apoptotic cells were seen having less green fluorescence along with diminished branches. On the contrary, PI dye started to penetrate from the ruptured membrane due to which there was increase in red fluorescence with increase in concentration and incubation time. At 40 µg/mL, there was complete loss of green fluorescence with only visualizing red fluorescence indicating the complete death of these cells. In the absence of staining, cells are not visible under fluorescence observation.
|
Reviewer 2 Report
The authors have developed a ten-membered peptide-drug library based on furane-tripeptide conjugates and extensively evaluated their bioactivity. The research is based on an interesting topic - selective tumour therapy with peptide-drug conjugates – and have promising results.
However, while reviewing the manuscript, the following observations were made:
- It is not clear to the reviewer why those amino acids (Nal, Phe, Ala) were chosen, there is no correct reference in the manuscript or extensive explanation. The manuscript is missing a sound hypothesis and the authors should correct this major flaw.
- The authors focused on three amino acids (Ala, Phe and Nal) to synthetise tripeptides. Therefore, it would seem logical, that the prepared library should consist of 27 conjugates, because 3*3*3 variations is possible. However, only 10 conjugates are presented in the manuscript. The authors should explain what logical considerations were taken into account that resulted in the “shortened” conjugate library.
- The IC50 values of the conjugates are presented as μg/μL. It is a minor flaw because the Mw of the conjugates are similar or nearly similar. But in order to correctly compare the biological effects, the IC50 values should be given in μM or mM.
The manuscript contains truly extensive biological research, and the results are also promising, but it has major flaws. Therefore, if the authors correct the errors and omissions mentioned above, I recommend this paper to be accepted.
Author Response
Dear Reviewer,
We thank you for your evaluation of our work and suggested corrections.
Please find below our answers.
regards
M Maresca
|
S. No. |
Comments |
Answers |
|
1 |
It is not clear to the reviewer why those amino acids (Nal, Phe, Ala) were chosen, there is no correct reference in the manuscript or extensive explanation. The manuscript is missing a sound hypothesis and the authors should correct this major flaw. |
Relevant references (ref. no. 24,25,26) are included along with explantation in the introduction. |
|
2 |
· The authors focused on three aminoacids (Ala, Phe and Nal) to synthetise tripeptides. Therefore, it would seem logical, that the prepared library should consist of 27 conjugates, because 3*3*3 variations is possible. However, only 10 conjugates are presented in the manuscript. The authors should explain what logical considerations were taken into account that resulted in the “shortened” conjugate library. |
There is a great interest in developoment of peptides of short sequences due to their easy design, synthesis and characterization, low cost, and most importantly their high biodegradability and biocompatibility. This has been included in manuscript with a reference ]20]
|
|
3 |
· The IC50 values of the conjugates are presented as μg/μL. It is a minor flaw because the Mw of the conjugates are similar or nearly similar. But in order to correctly compare the biologica leffects, the IC50 values should be given in μM or mM. |
IC50 in µM are now indicated in Table 1 of the revised version |
Reviewer 3 Report
In this manuscript, the authors developed furan-conjugated tripeptides as potent antitumor drugs. One of the conjugates, 4 showed specific antitumor activity against HeLa cells. Although the concept of furan-conjugated tripeptides was already reported by other groups, the information of the conjugates may be useful for the readers to design the new antitumor drugs. However, I strongly recommend to modify the following points.
- What is the reason of specific antitumor activity of compound 4 against HeLa cells? How about the selectivity of other conjugates? The authors should explain the expected reason of the cell selectivity in discussion.
- It is difficult to evaluate the mechanism of antitumor activity of compound 4 only by the AFM and PI staining experiments. Is there any specific target of compound 4 in cells? Does compound 4 disrupt the cell membrane or bind to intracellular targets? The authors should show the experimental evaluation of the mechanism and/or hypothesis of the antitumor activity of compound 4.
- Since compound 4 is L-peptide, it may have low proteolytic stability compared to D-peptide. The reviewer recommends to check the proteolytic stability of compound 4.
- Detailed anti-proliferative assay should be described in Materials and Methods section.
- Table 3 and 4 are too small to check. These tables should be modified.
Author Response
Dear Reviewer,
We thank you for your evaluation and comments on our manuscript.
We tried our best to answer all points.
Please find below
regards
M Maresca
|
S. No. |
Comments |
Answers |
|
1 |
What is the reason of specific antitumor activity of compound 4 against HeLa cells? How about the selectivity of other conjugates? The authors should explain the expected reason of the cells electivity in discussion. |
We have no explanation about the selectivity of conjugate 4 against HeLa cells. One can suggest that the specificity is related to a better and selective insertion of conjugate 4 into the membrane and mitochondria of HeLa cells compared to other cells tested. But this needs to be further evaluated through cell uptake assay. Similarly, we agree with reviewer that the action of other less active conjugates on different cell types needs to be explored. But we did not do it yet as the present study identified conjugate 4 as the more active conjugate. This has been indicated lines 349-350 |
|
2 |
It is difficult to evaluate the mechanism of antitumor activity of compound 4 only by the AFM and PI staining experiments. Is there any specific target of compound 4 in cells? Does compound 4 disrupt the cell membrane or bind to intracellular targets? The authors should show the experimental evaluation of the mechanism and/or hypothesis of the antitumor activity of compound 4. |
The AFM and the PI assay clearly showed that conjugate 4 acts through a membranolytic action. This means that the conjugate targets the membranes (plasma membrane and mitochondrial membrane), suggesting lipids are selective targets. Future experiments will be performed using liposomes and monolayer assay to confirm the interaction and insertion of conjugate 4 into lipids. This has been indicated lines 347-349 |
|
3 |
Since compound 4 is L-peptide, it may have low proteolytic stability compared to D-peptide. The reviewer recommends to check the proteolytic stability of compound 4. |
Although the fact the conjugate 4 is made of L-AA does not prevent it to be active in vitro, we agree that it may be sensitive to proteolysis. Futur studies will be conducted to evaluate the sensitivity of conjugate 4 to proteases and to overcome it if needed through the use of D-AA and / or vectorisation/encapsulation. |
|
4 |
Detailed anti-proliferative assays should be described in Materials and Methods section. |
Details have been indicated in the revised version (lines 177 to 188) |
|
5 |
Table 3 and 4 are too small to check. These tables should be modified. |
Table 3 and 4 are correctly presented in supplimentery material |
Round 2
Reviewer 1 Report
The manuscript has been revised according. It just needs typographical design.
Author Response
Thanks
Reviewer 2 Report
The authors did not provide satisfactory answers to the questions concerning the problem of the "shortened" peptide library and the selected amino acids. The article has not been improved on the subject of the questions either. It is the professional opinion of the reviewer's that mere references of review articles are no substitutes for complex and logical answers for such difficult questions.
In view of all this, I do not recommend the article to be published.
Author Response
Reviewer 2 indicated that « It is not clear to the reviewer why those amino acids (Nal, Phe, Ala) were chosen, there is no correct reference in the manuscript or extensive explanation. The manuscript is missing a sound hypothesis and the authors should correct this major flaw. » and that « The authors focused on three aminoacids (Ala, Phe and Nal) to synthetise tripeptides. Therefore, it would seem logical, that the prepared library should consist of 27 conjugates, because 3*3*3 variations is possible. However, only 10 conjugates are presented in the manuscript. The authors should explain what logical considerations were taken into account that resulted in the “shortened” conjugate library. »
Answer : We thanks Reviewer 2 for these comments. Although we did not mentioned it in the initial version of the manuscript, at the beginning of the study, 20-member furan conjugated tripeptide library was synthesized using Fmoc synthetic methodology on Rink amide resin in which each member had randomly selected L or D configured natural and unnatural amino acid residues using hydrophobic, hydrophilic, or cationic residues (Figure-S1 and Figure-1). This is now indicated in the new revised version of the manuscript (page 6, line 236-249). All compounds were fully characterized by mass spectrometry and detailed NMR spectroscopic studies (see supplementary data). Although not all combinations were synthetized, we believe having 20 versions allows already to do SAR study. Indeed, antiproliferative assay revealed that compounds having at least one residue in D-configuration were all inactive against HeLa cancer cell line (Figure S1, supporting information). These findings helped us to identify the shorter series of conjugate (1-10) of unnatural and natural amino acids in L-configuration with varied activity against HeLa cell lines and here we discussed SAR of anticancer conjugates in detail in this manuscript.
Reviewer 3 Report
The manuscript was improved by addressing the reviewer's comments (though not perfectly). The reviewer thinks that the current version is suitable for the publication.
Author Response
Thanks